# Anthropometric Parameters of Children with Congenital Zika Virus Exposure in the First Three Years of Life

**DOI:** 10.3390/v14050876

**Published:** 2022-04-23

**Authors:** Elisa Barroso de Aguiar, Sheila Moura Pone, Saint Clair dos Santos Gomes Junior, Fernanda Valente Mendes Soares, Andrea Araujo Zin, Zilton Farias Meira Vasconcelos, Carla Trevisan Martins Ribeiro, José Paulo Pereira Junior, Maria Elisabeth Lopes Moreira, Karin Nielsen-Saines, Marcos Vinicius da Silva Pone

**Affiliations:** 1National Institute of Women, Children and Adolescents’ Health Fernandes Figueira, Oswaldo Cruz Foundation, Rio de Janeiro 22250-020, Brazil; sheila.pone@iff.fiocruz.br (S.M.P.); scgomesjr@gmail.com (S.C.d.S.G.J.); fevalente@gmail.com (F.V.M.S.); zin.andrea@gmail.com (A.A.Z.); zvasconcelos@gmail.com (Z.F.M.V.); carla.ribeiro@iff.fiocruz.br (C.T.M.R.); josepaulo@globo.com (J.P.P.J.); bebeth@iff.fiocruz.br (M.E.L.M.); marcos.pone@iff.fiocruz.br (M.V.d.S.P.); 2David Geffen School of Medicine, University of California, Los Angeles, CA 90095, USA; knielsen@mednet.ucla.edu

**Keywords:** zika virus, zika virus infection, growth, malnutrition

## Abstract

Little is known about the impact of congenital Zika virus (ZIKV) exposure on growth in the first years of life. In this prospective cohort study,201 ZIKV antenatally-exposed children were followed at a tertiary referral center in Rio de Janeiro, Brazil. Eighty-seven were classified as congenital Zika syndrome (CZS) patients and 114 as not congenital Zika syndrome (NCZS); growth parameters were described and compared between groups and with WHO standard growth curves. Thirty-four (39%) newborns with CZS and seven (6%) NCZS were small for gestational age (*p* < 0.001). NCZS mean weight measures ranged from −0.45 ± 0.1 to 0.27 ± 0.2 standard deviations (SD) from the WHO growth curve median during follow-up, versus −1.84 ± 0.2 to −2.15 ± 0.2 SD for the CZS group (*p* < 0.001). Length mean z-scores varied from −0.3 ± 0.1 at 1 month to 0.17 ± 0.2 SD between 31 and 36 months in the NCZS group, versus −2.3 ± 0.3 to −2.0 ± 0.17 SD in the CZS group (*p* < 0.001). Weight/height (W/H) and BMI z-scores reached -1.45 ± 0.2 SD in CZS patients between 31 and 36 months, versus 0.23 ± 0.2 SD in the NCZS group (*p* < 0.01). Between 25 and 36 months of age, more than 50% of the 70 evaluated CZS children were below weight and height limits; 36 (37.1%) were below the W/H cut-off. Gastrostomy was performed in 23 (26%) children with CZS. During the first three years of life, CZS patients had severe and early growth deficits, while growth of NCZS children was normal by WHO standards.

## 1. Introduction

Zika virus (ZIKV) is an arbovirus detected in the Americas for the first time in 2015, during an exanthematous disease outbreak in Brazil [1,2,3]. In 2016, studies confirmed a causal relationship [4,5,6,7,8] between infection during pregnancy and occurrence of neurological [6,9,10,11], ocular [12,13,14] and osteoarticular damage [15,16] in the fetus.

Growth assessment is a critical part of pediatric health monitoring [17]; standards established by the World Health Organization (WHO) were adopted in at least 125 countries [18,19]. Children with cerebral palsy (CP) frequently present nutritional and growth disorders [20,21,22], and nutritional status correlates with secondary and/or chronic conditions and the use of health services [23]. Children with microcephaly, intracranial calcifications and neurological damage associated with ZIKV infection during pregnancy present with various levels of CP, but little is known about the impact of congenital ZIKV infection on growth [15,24,25,26,27,28,29,30].

The objective of this study was to describe growth parameters for weight, height and head circumference (HC) in children with congenital ZIKV exposure. Internationally accepted growth curves were used to compare the evolution of children with Congenital Zika Syndrome (CZS) with equally antenatally ZIKV-exposed children without clinical, radiological or ophthalmological alterations at birth (“not Congenital Zika Syndrome”—NCZS). Parameters which potentially impact growth were also evaluated.

## 2. Materials and Methods

This study was conducted at the Instituto Nacional de Saúde da Mulher, da Criança e do Adolescente Fernandes Figueira (IFF), in Rio de Janeiro, Brazil, a National Institute from the Oswaldo Cruz Foundation (FIOCRUZ) and the Brazilian Ministry of Health referral center for fetal malformations and pediatric infectious diseases. It was approved by the institutional review board at IFF and all guardians provided written informed consent.

Patients were referred to our Institute due to ZIKV maternal infection, intrauterine diagnosis of microcephaly or clinical and radiological presentation suggestive of CZS after birth.

### 2.1. Inclusion and Exclusion Criteria

All patients born to mothers with positive reverse transcription-polymerase chain reaction (RT-PCR) results in blood, urine, placenta or amniotic fluid samples, or with laboratory confirmation of fetal infection (positive RT-PCR or immunoglobulin M (IgM) in specimens of infant—blood, urine or cerebrospinal fluid) were eligible for the study.

Findings previously described by Moore et al. [16] as suggestive of ZIKV congenital infection (typical cranial morphology, brain and ocular anomalies, congenital contractures and neurologic sequelae) were used as criteria to define CZS in patients born during and in the aftermath of the ZIKV epidemic, regardless of laboratory confirmation. Infants with other congenital infections, genetic syndromes and prematurity (defined as gestational age (GA) at birth of below 37 weeks) were excluded.

ZIKV infection was diagnosed using RT-PCR assays with the QuantiTectProbe RT-PCR kit (manufactured by QIAgen, Maryland, USA), using primers and cycles described previously [31], after extraction of RNA with the TRIzol reagent, according to the manufacturer’s instructions. ZIKV serologic testing of infants was performed in duplicate serum aliquots using IgM antibody capture Zika MAC-ELISA from the Center for Disease Control and Prevention (CDC, Fort Collins, CO, EUA), according to the manufacturer’s instructions [32].

### 2.2. Study Procedures

We followed patients from March 2016 to December 2019. Pediatric infectious diseases experts documented clinical data; participants were evaluated by geneticists, neurologists, ophthalmologists, pediatric orthopedists, social workers and nutritionists.

Trimester of ZIKV infection during pregnancy was defined as the period in which the symptoms occurred. GA at birth was defined by Ballard’s assessment, maternal last menstruation date or ultrasonography. Z-scores for weight, height and HC at birth, according to GA and gender, were calculated by the International Newborn Size at Birth Standards application, developed by the Intergrowth 21^st^ Project [33]. Microcephaly was defined as HC below −2 (moderate) or −3 (severe) standard deviations (SD); small for gestational age (SGA) was defined as weight below −1.28 SD for GA and gender.

Neuroimaging (trans fontanelle ultrasound, computed tomography (CT) or nuclear magnetic resonance (MRI)) and ophthalmologic evaluations were performed in all participants. Diagnosis of arthrogryposis was defined by a pediatric orthopedist.

Neonatal complications included seizures, respiratory distress, jaundice, feeding difficulty or weight gain, hypoglycemia and sepsis. During follow-up, the main causes of hospitalization were urinary tract infections, respiratory diseases (pneumonia, bronchiolitis, bronchospasm), epilepsy and surgeries (mainly gastrostomy, ventriculoperitoneal shunt, herniorrhaphy, tracheostomy and orthopedic surgeries).

Three socioeconomic variables were assessed through interview at the first visit: maternal education, monthly family income and receipt of government aid. Maternal education was classified according to the Brazilian educational system: primary education (divided in 1–4 and 5–8 years of study); high school (9–11 years of study); and college. Monthly family income was divided according to the Brazilian Institute of Geography and Statistics (IBGE) classification [34]: class A (above 20 basic salaries (BS, equivalent to the minimum wage), B (10–20 BS), C (4–10 BS), D (2–4 BS) and E (less than two BS).

Patients were initially classified into four groups: NCZS and CZS with and without laboratory confirmation. Among the NCZS group, all mother–infant pairs had laboratory confirmation (of the mother, child or both) of ZIKV infection, and children had no clinical, radiological or ophthalmologic abnormalities suggesting congenital infection. Patients in the CZS groups had compatible findings of congenital infection, regardless of diagnosis of microcephaly. Children with CZS and positive RT-PCR (from the mother or child) or IgM were compared to those with CZS without laboratory confirmation; there was no significant difference between groups regarding demographic, clinical and socioeconomic data (Table A1). Averages of anthropometric measures from both groups were not statistically different in the analyzed periods (Appendix A). This analysis corroborated the previous clinical diagnosis, allowing classification as a single group for further comparisons.

Nutritionists measured weight, height and HC at each visit, using an electronic pediatric weighing-machine, a horizontal anthropometer and a non-extensible measuring tape. We scheduled monthly consultations from birth to six months, every three months up to two years and every semester between two and three years. When more than one measure was available, we calculated the mean value for the period. Means of all measures available in each group during each established period were calculated, and those averages were used to chart growth curves.

We used the WHO Anthro software (available at https://www.who.int/childgrowth/software/en/, accessed on 1 May 2020) to calculate anthropometric indicators: z-scores for weight (W/A), height (H/A), weight for height (W/H), HC (HC/A) and body mass index (BMI/A) for age and gender. Critical values defined by the WHO for nutritional diagnosis were used [18] and malnutrition was defined according to the W/H classification. Z-scores below −2 and above −3 SD from the WHO Child Growth Standards median define children as underweight, stunted and wasted; children below −3 SD had very low weight, very short stature and severe wasting or malnutrition [35].

### 2.3. Statistical Analysis

Statistical analysis was performed using SPSS software 21.0. We evaluated potential associations between categorical variables using Chi-square test; *t*-student and ANOVA tests were used for numerical variables. Two-sided *p*≤ 0.05 was considered to be statistically significant.

## 3. Results

After applying exclusion criteria, 201 of 246 referred mother–infant pairs were followed in this study; 114 children were classified as NCZS and 87 as CZS. Maternal or fetal infection was confirmed by RT-PCR or IgM in 43 (49.4%) patients in the CZS group and in all NCZS patients (Figure 1).

Table 1 compares demographic, socioeconomic, gestational and clinical characteristics in each group. The majority of patients from the CZS group had microcephaly at birth (77%). Although all infants in the NCZS had a history of maternal rash during pregnancy (100%), compared to 78.2% of infants with CZS, 67.8% of mothers from the CZS group reported rash in the first trimester, versus 24.6% of the NCZS group (*p* < 0.01).

Two patients with CZS had normal neuroimaging, but compatible ophthalmologic abnormalities (microcornea, coloboma and microphthalmia) and laboratory confirmation of maternal infection. One patient in the NCZS group had microcephaly at birth (HC: 32 cm, z-score: −2.1 SD), with normal head CT and fundoscopy, and HC between zero and one SD in the WHO charts in subsequent visits. Three patients classified as NCZS had an abnormal initial neuroimaging and one had an abnormal fundoscopy, which were either suggestive of other etiologies or restricted to minor abnormalities, absent in subsequent neuroimaging exams.

Thirty-four (39%) patients with CZS were SGA, versus seven (6%) of the NCZS group (*p* < 0.01); neonatal complications were significantly more frequent in the CZS group (31 cases, 35.6%) than in the NCZS group (16 cases, 14%, *p* < 0.001).

Twenty-seven (32%) patients with CZS had at least one urinary tract infection treated in the outpatient setting, compared to seven (8%) in the NCZS group (*p* < 0.001). A total of 83.7% of children from the CZS group (versus three (3%) NCZS children) needed anticonvulsant drugs (*p* < 0.001). Twenty-eight CZS patients had refractory epilepsy (data not shown).

Children with CZS were more frequently admitted to a hospital during the first three years of life because of any complications (mainly respiratory disorders, urinary tract infection, seizures and surgeries). A total of 75.5% of CZS patients were hospitalized during follow-up (versus 22.5% of NCZS, *p* < 0.001). Five children (2.5%) died during follow-up, all with CZS.

Gastrostomy was performed in 23 children (26.4%) with CZS, with a median age of 14.5 (0–36) months, and none of the NCZS group (*p* < 0.001). Frequency of breastfeeding was similar between the two groups (*p* = 0.3) and around 90%; however, mean duration of breastfeeding was significantly longer in NCZS children (10.4 months versus 7.1 months, *p* = 0.02), as was the time of exclusive breastfeeding: 3.7 versus 2.7 months, *p* = 0.001 (data not shown).

Family income was significantly different: 63% of the families from the CZS group were classified as class E, and 27% as class D; in the NCZS, 40% were in class E and 30% in class D (*p* = 0.008). Twenty-three percent of CZS patients’ families received government aid, versus 8.6% of NCZS (*p* = 0.001).

Median age at first evaluation was one month for both groups. Measures at birth were available for all patients; in each established period, we evaluated on average 59 (51%) NCZS patients (32–69 patients) and 58 (66%) CZS patients (40–69 patients).

At birth, the average measurements obtained from patients with CZS (weight: 2796 ± 47g, height: 47.4 ± 0.3 cm, and HC: 30 ± 0.3 cm) and their respective mean z-scores (−0.93 ± 0.12, −0.86 ± 0.16 and −2.75 ± 0.2 SD) were significantly lower than in the NCZS group (weight: 3300 ± 43g, height: 49.3 ± 0.2 cm; HC: 34.9 ± 0.12 cm, corresponding to 0.27 ± 0.1, 0.18 ± 0.1 and 0.98 ± 0.1 SD, *p* < 0.001). Disparity between the groups increased during follow-up, maintaining a statistically significant difference for both absolute measures and SD from the WHO Growth Chart Standards (Figure 2A–D).

Average measures of children in the NCZS group remained within the normal range and close to zero SD from the WHO Growth Standards for all evaluable parameters. In the first month, the mean weight of patients with CZS was already 1.84 ± 0.2 SD below the WHO median, versus −0.45 ± 0.1 SD in the NCZS group (*p* < 0.001). Between 31 and 36 months, the mean weight of CZS patients was 2.15 ± 0.19 SD below the WHO median and 0.27 ± 0.2 SD in the NCZS group (*p* < 0.001) (Figure 2B).

The mean height z-score had similar patterns: −2.3 ± 0.3 SD in the CZS group in the first month, versus −0.3 ± 0.1 SD in the NCZS group (*p* < 0.001); and −2.0 ± 0.17 SD in the CZS group between 31 and 36 months, versus 0.17 ± 0.2 SD in the NCZS group, *p* < 0.001 (Figure 2D).

Weight/height and BMI z-scores were significantly lower in patients with CZS after four months of age. In the first month, patients with CZS had a mean W/H z-score of 0.2 ± 0.2 SD, versus −0.1 ± 0.15 SD in the NCZS, reaching −1.45 ± 0.2 SD in the CZS between 31 and 36 months and 0.23 ± 0.2 SD in the NCZS group (*p* < 0.001) (Figure 3A).

In the third year of life, 37 (52.8%) of the 70 evaluated CZS children were below the established limits for weight/age standards, and 41 (51.7%) were classified as stunted or severely stunted. Eight patients (11.4%) had severe malnutrition and eighteen (25.7%) were wasted (Table 2). A total of 83 (41.3%) patients were lost to follow-up, 66 of them (79.5%) in the NCZS group.

Patients with CZS maintained a severe HC impairment, with a mean z-score below −4 SD throughout follow-up and reaching −5.2 ± 0.3 SD between 31 and 36 months (versus 0.66 ± 0.18 SD in the NCZS group (*p* < 0.001)), when 44 (84.6%) of those evaluated presented severe microcephaly. At this age, mean HC was 41.5 ± 0.45 cm for the CZS patients and 49.7 ± 0.3 cm for NCZS, *p* < 0.001 (Figure 3C,D).

## 4. Discussion

This study described the growth of ZIKV-exposed children evaluated in a tertiary hospital, based on the WHO Growth Standards and comparing groups defined as CZS and NCZS. We evaluated 201 children, with an average of 51 measurements in each group throughout the study period. Previous studies evaluating growth in this population included smaller populations, analyzed at a single moment in time or with shorter follow-up periods [24,25,26,27,36].

Similarities between the children with CZS, with and without laboratory confirmation of ZIKV exposure, reinforced the value of a clinical diagnosis. This is an important finding, given the severe limitations precluding laboratory confirmation of ZIKV congenital infection, including the short window of positivity for RT-PCR assays and IgM cross-reaction with responses to other flaviviruses [37].

In this study, we found a high frequency of SGA infants in the CZS group when compared to the NCZS group (39% versus 6%, *p* < 0.001). Animal models and case series have shown an increased frequency of intrauterine growth restriction (IUGR) and low birth weight in the fetuses of mothers infected with ZIKV [28,38,39,40,41], affecting up to 18–28% of these pregnancies [36,38,42] and possibly associated with placental damage [39].

Although previous studies suggest that infants with in utero ZIKV exposure but without CZS are at risk of abnormal neurodevelopmental outcomes [43,44,45], our findings suggest that physical growth was not compromised in this group. Mean measures of weight, height and HC in the NCZS group of children were close to zero SD when compared to WHO standards, while CZS patients presented early and significant impairment in postnatal growth.

At birth, only the mean HC was below −2 SD in the CZS group; however, all measures were significantly lower than in the NCZS group. Throughout follow-up, there was no recovery of cephalic growth, and mean HC remained over four to five SD below average. Microcephaly at birth has been a hallmark of CZS since its first descriptions [4,6,7,8,9]. Reinforcing our findings, previous studies also found impaired postnatal cephalic growth in these children, including patients with clinical and radiological findings of CZS but without microcephaly at birth [30].

The CZS group had significantly lower mean z-scores for weight and height than the NCZS group, reaching a difference of up to 2 SD; by the end of follow-up, the CZS mean measures approached −2 SD. This group also showed progressive reduction in the W/A z-score—the most important parameter for assessing acute malnutrition, according to the WHO [18]. Corroborating our findings, animal models also suggested limitation of postnatal growth after congenital infection with zika virus [46,47], especially when infection occurred in the first two trimesters of pregnancy [47].

In a previous case series, with a smaller population and shorter follow-up, França et al. found adequate weight and height measures at birth in eight children with CZS, with weight compromise at 20.5 months [26]. Moura et al. showed impaired weight, height and HC growth of 48 children with CZS until eight months [24]. Two cross-sectional studies, evaluating 19 and 46 children with CZS found similar results [27,36]. In a series of 29 cases, Prata-Barbosa et al. suggested an association between neurological impairment and weight–height deficit in children exposed to ZIKV during the first two years [25].

In the third year of life, 52.8% of our patients with CZS were underweight and 57.1% had stunting. Wasting was detected in 26 (37.1%), eight of them classified as severe. Dos Santos et al. also found that W/H was proportional in more than 60% of 65 infants with microcephaly at 13–24 months, which highlights an anthropometric profile in which both weight and height are affected [48].

In our study, 26% of children with CZS required gastrostomy for G-tube feeding due to severe malnutrition, dysphagia, feeding difficulties and risk of broncho-aspiration. Previous studies diagnosed oropharyngeal dysphagia in as many as 79% of children with CZS [49]; in one case series, 10 of 28 patients (35.7%) required G-tubes until 24 months [50]. Dysphagia in these patients can be severe and associated with lack of oral and upper respiratory tract sensitivity, increasing the risk for aspiration, particularly of liquid foods [29]. A small case series of postmortem analysis suggested that lung disease associated with pulmonary aspiration could be an important cause of death among infants with CZS [51].

Breastfeeding frequency was similar between groups, but duration was lower in the CZS group; previous studies suggested impaired suction in infants with CZS after the first months, when the suction reflex disappears, and early weaning in infants with microcephaly due to CZS has been described [29,48].

Defining adequate growth in a population with neurological impairment is challenging [23]. Growth curves for children older than two years with cerebral palsy have been proposed [52], but its applicability is questionable [53].

Growth deficit, especially in children with neurological impairment, is a multifactorial and complex process, related to nutritional, socioeconomic, endocrine and secondary conditions [20,54]. In this study, neonatal complications, hospital admissions and urinary tract infections were significantly more frequent in children with CZS. Nutritional impairment, together with other factors such as dysphagia and neurogenic bladder [55,56], may contribute to the occurrence of such complications; at the same time, their occurrence may contribute to the observed growth deficit.

Children with CZS frequently used more than one anticonvulsant drug. Epilepsy often makes feeding even more difficult; at the same time, chronic use of some antiepileptic drugs can promote gain or loss of weight [57,58].

In our study, socioeconomic factors were evaluated at a single moment; children with CZS were more frequently from very low-income families (class E). However, our data are limited for further analysis of the complex relation between socioeconomic factors and their impacts.

Our study has some limitations, and results cannot be extrapolated to the general population or individual ZIKV cases. Anthropometric data were not analyzed individually, our population stems from a referral system to a tertiary hospital and there was loss to follow-up over time, especially in the NCZS group. Feeding practices and food inadequacies, an important aspect in growth evaluation [48], were not routinely assessed.

Despite inherent difficulties of growth assessment in this population, our data are unique because we longitudinally describe a large number of ZIKV-exposed patients, most of them with laboratory confirmation of antenatal ZIKV infection. The significant impact observed in weight, height and W/H parameters consistently confirms the impairment in weight–height growth in children with CZS; growth deficit seems to be present since birth and worsens over time during the first three years of life.

## 5. Conclusions

Follow-up of approximately two hundred children with congenital exposure to ZIKV showed serious and early growth impairment among patients with CZS. Longitudinal anthropometric assessment of this large number of children is unprecedented in the literature. Growth impairment in CZS is probably influenced by multiple factors and our findings reinforce the need for a careful approach and further research, to avoid serious growth deficits and their impact on the overall health of these vulnerable children.

## Figures and Tables

**Figure 1 viruses-14-00876-f001:**
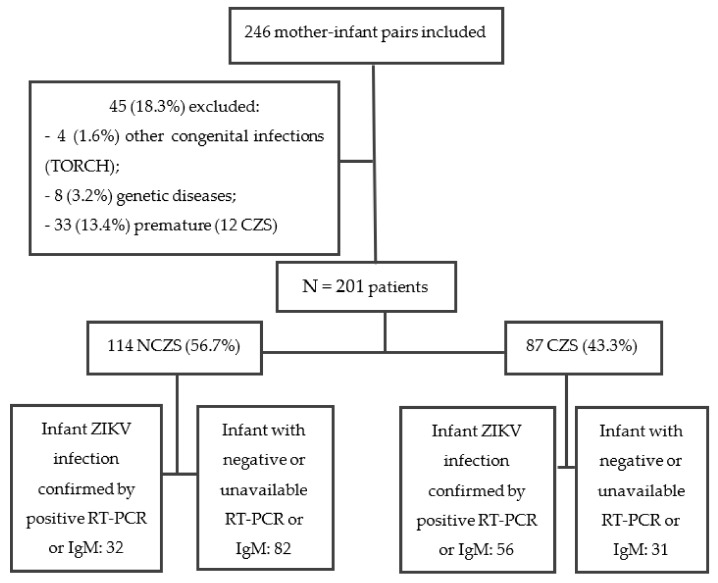
Flowchart of Participant Recruitment. ZIKV: zika virus; TORCH: toxoplasmosis, rubella, cytomegalovirus, herpes and others (HIV, syphilis); CZS: congenital zika syndrome; NCZS: not congenital zika syndrome; RT-PCR: reverse transcriptase-polymerase chain reaction; IgM: immunoglobulin M.

**Figure 2 viruses-14-00876-f002:**
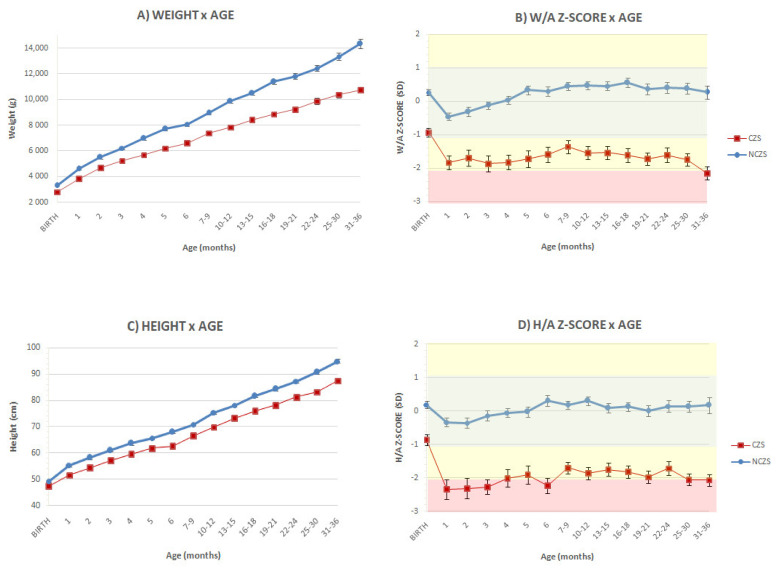
Weight and height absolute values and z-score means for age follow-up: CZS versus NCZS. Graphs compare (**A**) weight for age, (**B**) weight for age z-scores, (**C**) height for age and (**D**) height for age z-scores, based on WHO Growth Standards. W/A represents weight for age; H/A represents height for age; CZS: congenital zika syndrome; NCZS: not congenital zika syndrome. Bars represent standard error.

**Figure 3 viruses-14-00876-f003:**
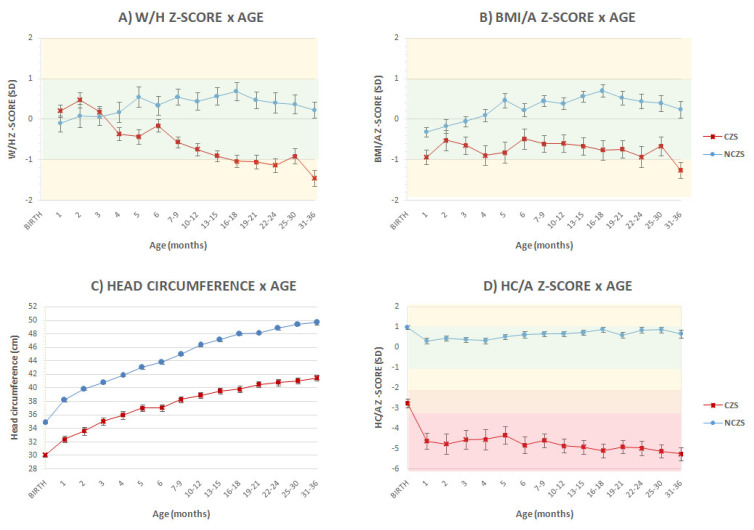
Head circumference, weight/height and BMI z-score means follow-up: CZS versus NCZS. Graphs compare (**A**) weight/height z-score for age, (**B**) body mass index/age z-score for age, (**C**) head circumference for age and (**D**) HC z-scores for age, based on the WHO Growth Standards. HC represents head circumference; W/H represents weight for height; BMI/A represents body mass index for age; CZS: congenital zika syndrome; NCZS: not congenital zika syndrome. Bars represent standard error.

**Table 1 viruses-14-00876-t001:** Clinical and socioeconomic characteristics and follow-up complications of children with CZS and NCZS.

			Total	Participants (*N* = 201)		*p*-Value
				NCZS (*N* = 114)	CZS (*N* = 87)	
				*N* (%)	*N* (%)	
**Sex**	Fem		104 (51.7%)	59 (51.8%)	45 (51.7%)	1.00
	Male		97 (48.3%)	55 (48.2%)	42 (48.3%)	
**Rash by trimester of pregnancy**	Absent		19 (9.5%)	0 (0%)	19 (21.8%)	<0.001
	1st		87 (43.3%)	28 (24.6%)	59 (67.8%)	
	2nd		63 (31.3%)	56 (49.1%)	7 (8%)	
	3rd		32 (15.9%)	30 (26.3%)	2 (2.3%)	
**Head circumference at birth** **(*n* = 201)**	Normocephaly		133 (66.2%)	113 (99.1%)	20 (23%)	<0.001
	Microcephaly (<−2 SD)		68 (33.8%)	1 (0.9%)	67 (77%)	
	Severe microcephaly (<−3 SD)		52 (25.9%)	0 (0%)	52 (59.8%)	
**Abnormal neuroimaging** **(*n* = 199)**	Yes		87(43.7%)	3 (2.7%)	85 (97.7%)	<0.001
	Not performed		2	2	0	
**Abnormal ophthalmologic evaluation** **(*n* = 199)**	Yes		51 (25.6%)	1 (0.9%)	50 (57.5%)	<0.001
	Not performed		2	2	0	
**Arthrogryposis (*n* = 201)**	Yes		15 (7.5%)	0 (0%)	15 (17.2%)	<0.001
**Small for Gestational Age (*n* = 201)**	Yes		41 (20.4%)	7 (6.1%)	34 (39.1%)	<0.001
**Neonatal complication** **(*n* = 201)**	Yes		47 (23.4%)	16 (14%)	31 (35.6%)	<0.001
**Breastfeeding** **(*n* = 176)**	Yes		159 (90.3%)	87 (92.6%)	72 (87.8%)	0.287
	Missing		25	20	5	
**Gastrostomy** **(*n* = 201)**	Yes		23 (11.4%)	0 (0%)	23 (26.4%)	<0.001
**Hospitalization** **(*n*= 179)**	All causes	Yes	86 (48%)	21 (22.5%)	65 (75.5%)	<0.001
		Missing	22	21	1	
	Urinary tract infection	Yes	21 (12.3%)	2 (2.3%)	19 (22.6%)	0.002
		Missing	30	27	3	
	Respiratory disease	Yes	46 (26.5%)	4 (4.6%)	42 (50%)	<0.001
		Missing	30	27	3	
	Epilepsy	Yes	22 (10.9%)	2 (2.2%)	20 (23.8%)	<0.001
		Missing	30	27	3	
	Surgery	Yes	35 (17.4%)	3 (3.5%)	32 (38.1%)	<0.001
		Missing	31	28	3	
**Anticonvulsant drugs** **(*n*= 181)**	Yes		75 (41.4%)	3 (3.2%)	72 (83.7%)	<0.001
	Missing		20	19	1	
**Urinary tract infection (outpatient)** **(*n* = 171)**	Yes		34 (19.9%)	7 (8%)	27 (32.2%)	0.003
	Missing		30	27	3	
**Death** **(*n* = 201)**	Yes		5 (2.5%)	0 (0%)	5 (5.7%)	0.010
**Maternal education** **(*n* = 171)**	1–4 years		9 (5.3%)	1 (1.1%)	8 (9.5%)	<0.001
	5–8 years		29 (17%)	11 (12.6%)	18 (21.4%)	
	High school		90 (52.6%)	44 (50.6%)	46 (54.8%)	
	College		43 (25.1%)	31 (35.6%)	12 (14.3%)	
	Missing		30	27	3	
**Government aid** **(*n*= 162)**	Yes		26 (16%)	7 (8.6%)	19 (23.5%)	0.001
	Missing		39	33	6	
**Family income classification** **(*n* = 179)**	A		4 (2.2%)	3 (3.1%)	1 (1.2%)	0.008
	B		3 (1.7%)	2 (2.1%)	1 (1.2%)	
	C		29 (16.2%)	23 (24.2%)	6 (7.1%)	
	D		52 (29.1%)	29 (30.5%)	23 (27.4%)	
	E		91 (50.8%)	38 (40%)	53 (63.1%)	
	Missing		22	19	3	

NCZS represents not congenital zika syndrome; CZS: congenital zika syndrome. Family income classification, according to IBGE: A = above 20 basic salaries (BS); B = 10–20 BS; C = 4–10 BS; D = 2–4 BS; E = less than 2 BS.

**Table 2 viruses-14-00876-t002:** Frequency of underweight status, stunting, wasting and microcephaly at 25–36 months in CZS versus NCZS patients.

	NCZS (*n* = 48)	CZS (*n* = 70)
**W/A< −3 SD (severely underweight)**	0 (0%)	22 (31.4%)
**W/A< −2 AND >−3 SD (underweight)**	0 (0%)	15 (21.4%)
**Total W/A < −2 SD**	0 (0%)	37 (52.8%)
**H/A < −3 SD (severely stunted)**	1 (2.1%)	14 (20%)
**H/A< −2 AND > −3 SD (stunted)**	1 (2.1%)	26 (37.1%)
**Total H/A < −2 SD**	2 (4.2%)	40 (57.1%)
**W/H< −3 SD (severely wasted)**	0 (0%)	8 (11.4%)
**W/H< −2 E > −3 SD (wasted)**	0 (0%)	18 (25.7%)
**Total W/H < −2 SD**	0 (0%)	26 (37.1%)
**HC/A< −3 SD (severe microcephaly)**	0 (0%)	57 (81.4%)
**HC/A < −2 E > −3 SD (microcephaly)**	0 (0%)	4 (5.7%)
**Total HC/A < −2 SD**	0 (0%)	61 (87.1%)

W/A: weight for age; H/A: height for age; W/H: weight for height; HC/A: head circumference for age; CZS: congenital zika syndrome; NCZS: not congenital zika syndrome.

## Data Availability

Not applicable.

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
