# Peer review of "Anthropometric Parameters of Children with Congenital Zika Virus Exposure in the First Three Years of Life"

_viruses, 2022, doi:10.3390/v14050876_

Round 1
Reviewer 1 Report
Thank you for the opportunity to read this manuscript. It was well written and an important study showing growth trajectories of in utero Zika exposed children both with and without CZS. I have no major comments.
A few minor comments:
- Abstract- For the SGA patients, would specify at birth or use the word newborns instead of patients to make clear that is referring to a birth measurement. Sentence on length mean z-scores should have over what time period.
- Introduction- 1st paragraph- would specify that the causal relationship between infection in pregnancy and occurrence of the conditions is in the fetus or offspring.
- Introduction- 2nd paragraph- The link to cerebral palsy should be defined. Have children with CZS been diagnosed with CP? Are children with CZS comparable to children with CP and in what ways. The inclusion of this seems to fit but needs to be better introduced into the context of Zika.
- Methods- define criteria used for CZS. This may fit in the inclusion paragraph. Also, in the inclusion paragraph the authors use the term liquor. I think a more common term would be cerebrospinal fluid so may consider changing the term used there.
- Figure 1 legend; CZS abbreviation is written as CSZ.
- Table 1. Title includes “complications of infants”, should this be “children” since over age 12 months?
- Table 1: I would prefer to see the microcephaly numbers presented to include normocephaly, microcephaly, and severe microcephaly, thus accounting for HC in all participants.
- Table 1: Neonatal complications is included here. What these complications were considered as should be included in the methods. Similarly, it would be interesting to know what surgeries the children needed. Is gastrostomy considered separately from these other surgeries?
- Results: Over what time period did the children need hospitalizations?
- Discussion: 5th paragraph, “lower than in the NCZS group, approaching -1SD.” I think the sentence is confusing as written and would be better simply as “lower than in the NCZS group.”
- Discussion line 372. Better to state “infants with CZS” instead of CZS infants.
- Discussion line 374 “impaired suction in children”, this should probably be “impaired suction in infants”
- Discussion- The socioeconomic factors and how this was different between the children with CZS and NCZS was very interesting. Can the authors provide a little more thought on why this may be?
Reviewer 2 Report
After reading the manuscript, it is inevitable to wonder about a possible cause/effect relationship between the family's economic situation and the association with the congenital Zika virus syndrome. Was the family's economic status assessed only once or was it also monitored as the child grew? This is because the reader can imagine that the child's condition can impoverish the family over time. Reflections on this were not sufficiently explored in the analysis of the results, although it is cited as a limitation of the study.
